# Microbial Diversity Associated with Gwell, a Traditional French Mesophilic Fermented Milk Inoculated with a Natural Starter

**DOI:** 10.3390/microorganisms8070982

**Published:** 2020-06-30

**Authors:** Lucas von Gastrow, Marie-Noëlle Madec, Victoria Chuat, Stanislas Lubac, Clémence Morinière, Sébastien Lé, Sylvain Santoni, Delphine Sicard, Florence Valence

**Affiliations:** 1STLO, INRAE, Institut Agro, 35042 Rennes CEDEX, France; Lucas.Von-Gastrow@inrae.fr (L.v.G.); marie-noelle.madec@inrae.fr (M.-N.M.); victoria.chuat@inrae.fr (V.C.); 2SPO, University of Montpellier, INRAE, Montpellier SupAgro, 34000 Montpellier, France; delphine.sicard@inrae.fr; 3UBPN, Maison de l’agriculture, GIE Elevages de Bretagne, Rond-Point Maurice Le Lannou, 35042 Rennes CEDEX, France; stanislas.lubac@gmail.com (S.L.); c.moriniere@gie-elevages-bretagne.fr (C.M.); 4Statistics and Computer Science Department, Univ Rennes, Agrocampus Ouest, CNRS, IRMAR - UMR 6625, F-35000 Rennes, France; sebastien.le@agrocampus-ouest.fr; 5AGAP, University of Montpellier, CIRAD, INRAE, Montpellier SupAgro, 34000 Montpellier, France; sylvain.santoni@inrae.fr

**Keywords:** fermented milk, lactic acid bacteria, sensory evaluation, *Lactococcus*, microbial community, back-slopping

## Abstract

Gwell is a traditional mesophilic fermented milk from the Brittany region of France. The fermentation process is based on a back-slopping method. The starter is made from a portion of the previous Gwell production, so that Gwell is both the starter and final product for consumption. In a participatory research framework involving 13 producers, Gwell was characterized from both the sensory and microbial points of view and was defined by its tangy taste and smooth and dense texture. The microbial community of typical Gwell samples was studied using both culture-dependent and culture-independent approaches. *Lactococcus lactis* was systematically identified in Gwell, being represented by both subspecies *cremoris* and *lactis* biovar *diacetylactis* which were always associated. *Geotrichum candidum* was also found in all the samples. The microbial composition was confirmed by 16S and ITS2 metabarcoding analysis. We were able to reconstruct the history of Gwell exchanges between producers, and thus obtained the genealogy of the samples we analyzed. The samples clustered in two groups which were also differentiated by their microbial composition, and notably by the presence or absence of yeasts identified as *Kazachstania servazii* and *Streptococcus* species.

## 1. Introduction

Within the huge diversity of fermented milk products, the production of some is based on the use of a natural starter and more particularly related to a specific fermentation practice called back-slopping. Back-slopping is characterized by its simplicity as it only requires a raw material to be transformed and a previous production for inoculation [1]. Because it offers a more reproducible and safer method compared to spontaneous fermentation, back-slopping is used for numerous traditional fermented foods, which include raw milk cheeses, natural wines, sourdough bread or lambic beers [2,3,4,5,6,7]. This method can also be regarded as an intermediate between spontaneous indigenous fermentation and fermentation controlled by selected pure cultures. Spontaneous fermentation can produce variable results because the fermentation process relies on endogenous microflora in the raw materials and environment and therefore on an undefined microbial consortium that may affect the repeatability and stability of the final products. In this context, to improve the control of production, particularly those at a large scale, standardized starters are increasingly being used. Although the use of standardized and selected starters can ensure the reproducibility of production, it also leads to a standardization of the products, with generally a small set of very similar strains being commercialized and used [8]. By contrast, back-slopping enables expression of the typicality associated with the raw material and transformation process but in a controlled manner, the limitation being that in some cases a drift of the final product over time may lead to production defects [7]. Gwell is a specific type of gros-lait, a traditional mesophilic fermented milk originating from the French region of Brittany. Gros-lait was very popular in the early 20th century and was almost only produced at home for self-consumption. Its production decreased gradually until the middle of the 20th century, in parallel with the decline of Bretonne Pie Noir [9], an iconic cattle breed from Brittany, which was replaced by more productive breeds such as Prim’Holstein. The Bretonne Pie Noir breed was almost extinct until a rescue plan was initiated in 1976. Since then, headcounts have increased but are still lower than the threshold below which extinction is no longer a threat. Farmers registered the Gwell trademark in 1993 as designating a gros-lait made of milk from Bretonne Pie Noir cows. By linking the product to the breed, they dedicated their efforts to rescuing both a local cattle breed and a traditional product [10]. As part of their efforts to develop the product, two samples were sent by a producer to a private laboratory for microbiological analysis and *Lactococcus lactis* was identified as the main species. The Gwell making process was quite diverse until the 2000s, but it shared at least two main characteristics: use of the back-slopping method, where a batch of a previous Gwell production is used to inoculate thermized milk, and mesophilic fermentation (where the temperature of the fermentation process enables the development of mesophilic microorganisms, i.e., 20–30 °C). Working together, Gwell producers specified the recipe in order to obtain a more recognizable product, and specifications were compiled in 2014. A typical production process consists in pasteurizing the milk to 85 °C, cooling it to 35 °C, adding 5% to 10% (*v*/*v*) cold Gwell and homogenizing the mix, and then incubating at 30 °C for 3.5 to 6 h. As a result of fermentation by mesophilic lactic flora, the lactose is metabolized into lactic acid that causes a drop in the pH and concomitant clotting of the milk. The resulting curd is then stored at 4 °C before consumption, and it can be used as a ferment for the next production. Despite all the precautions taken, the Gwell can sometimes change quite drastically, leading to a loss of the product. The most frequent problems are the slowing down of acidification and texture or taste defects. The producers then have no choice but to obtain a new Gwell from another producer, which therefore hampers Gwell-driven production. However, these manufacturing accidents offer an interesting model for microbiologists to study the dispersion and resilience of microbial communities. Various mesophilic fermented milks have been described among dairy products [3] and include Vilii in Finland, Dahi in India or Dadih in Indonesia, whose process is similar to that of Gwell [11]. They are obtained after a thermization step of the milk, by inoculation of a previous day production. The more important variations between their processes concern the origin of the milk according to the considered country (cow, buffalo or even yak), and/or the temperature of the thermization step and/or and the duration and temperature of incubation during the fermentation step. As Dahi and Dadhi are mainly produced at domestic scale, the detailed process is very hard to determine, in general the incubation is performed at room temperature, the milk is boiled from several minutes to one hour, then cooled before the inoculation step [12]. For Viili, semi industrial scale production mentions a pasteurization step and an incubation step of 20 h at 20 °C. The Viili ecosystem is dominated by lactic acid bacteria (LAB) and more particularly *Lactococcus lactis* and *Leuconostoc mesenteroides* [13,14]. For Dahi, the bacterial ecosystem comprises several lactic genera: *Streptococcus*, *Lactobacillus*, *Enterococcus*, *Leuconostoc*, *Lactococcus* and *Pediococcus* [15], while for Dadih the bacterial ecosystem is dominated by LAB and more particularly *Lactococcus lactis* species [16]. In the traditional French cheese called Pelardon, made from raw goat milk, producers use a natural starter. The milk is indeed inoculated with whey from the previous production and back-slopping practices are also employed. Coagulation is conducted at a low temperature (20 °C), which is quite similar to Gwell. Tormo et al. [5] characterized the bacterial community in these traditional raw goat cheeses using culture-dependent methods and showed that the community is largely dominated by *Lactococcus lactis* species, accounting for 60% of the strains isolated. Traditional raw goat cheese producers sometimes have to deal with a drift of their production related to the decline in the activity of their natural starter, which has the same defects as Gwell [17]. In this context, Gwell offers an interesting model to study the dynamics of the microbial community associated with back-slopping practices and their associated defects. This study aims at better understanding the Gwell ecosystem, by characterizing in detail the microbial community and analyzing some factors influencing Gwell microbial composition. That would allow the producers to better control the Gwell ecosystem and to avoid Gwell loss accidents. Thus, in the context of a participatory research framework involving almost all Gwell producers (13 out of 15 at the time of the study), we characterized Gwell from both an organoleptic point of view and in terms of its microbial community, using culture-dependent and –independent methods, following a two-step approach. First, in a selected panel of Gwell samples, we characterized the typicality of Gwell from an organoleptic point of view. We then selected typical Gwell samples in order to isolate and identify the associated microbial community. In a second step, we used metabarcoding to analyze the microbial community in a larger panel of samples so as to obtain an overview of the microbial diversity associated with Gwell exchanges between producers.

## 2. Materials and Methods

### 2.1. Gwell Sampling and Data Collection

Gwell samples were collected at two different time points with an 18-month interval between them (May 2017 and November 2018). The first sample batch (denoted “a”) was supplied by the producers for sensory analysis and the samples were then used for culture-dependent microbial characterization. The second sample batch (denoted “b”) was sent by courier and received within 24 h. These samples were used for metabarcoding analysis. The data concerning these samples, and notably Gwell exchanges between producers, were collected using a survey sent in with the samples. These data were completed with semi-structured interviews carried out separately in each farm where we also followed Gwell production. The list of Gwell samples and the respective analyses performed are shown in Table 1.

### 2.2. Sensory Evaluation of Gwell

#### 2.2.1. Two-Steps Sensory Analysis

In order to define a typical Gwell, generate sensory descriptors and group together Gwell samples perceived as being similar, we performed a two-step sensory analysis. A Gwell was considered to be typical if it was recognized by the producers as being representative of the product per se. During the first step, an expert panel of 15 judges analyzed 10 different Gwell samples. These judges all knew the product and were used to consuming it, representing almost all Gwell producers at the time of the study. A random three-digit code was assigned to each sample. They were anonymized and duplicated in order to quantify the reliability of each panelist’s judgment. Tasting samples were given in a random order and the jury was asked to decide, according to their knowledge of the product, whether they were a typical Gwell or not, taking into account of their organoleptic characteristics. After grouping the samples, they described the common organoleptic properties of both typical and non-typical Gwell. The second step consisted in a Gwell sorting task followed by a verbalization task carried out by a panel of 29 untrained judges who did not necessarily know the product. This second step was carried out four days after the first step using Gwell samples from the same batch. However, two of the ten previously analyzed Gwell samples were discarded: GW11a because coliforms were detected at excessive levels and GW2a because an insufficient quantity was available, so eight samples remained. A random three-digit code was assigned to each sample. Panelists were asked to group together the samples they perceived as being the most similar, taking account of the characteristics (texture, odor and/or taste) they considered as being important to differentiate the products, according to the method described by Cadoret et al. (2009). Once the groups had been produced, the panelists were asked to associate specific descriptors with each group (verbalization task). To ensure that the panelists were able to group similar samples, one sample was tested in duplicate.

#### 2.2.2. Statistical Analysis

All statistical analyses were performed using R 3.6.2. For the first step, we calculated the percentage of times a sample was considered to be typical by the expert panel. Samples that obtained a percentage higher than 50% were considered to be “typical Gwell” that could serve as a reference to describe the product. In order to assess the reliability of these judgements, we also calculated the percentage of times each judge assigned the duplicated samples to the same category. To generate descriptors to characterize Gwell, frequency analysis was performed using the R [18] “FactoMineR” [19] package on the words used by the tasters to describe the Gwell they judged to be typical and non-typical. These descriptors were cleaned and standardized without interpreting their meaning. A graphic representation was made using “Wordle” online software. We then performed a Multiple Correspondence Analysis (MCA) on the results of the second step using the R “FactoMineR” and “factoextra” [20] packages.

### 2.3. Microbial Enumeration

Enumeration was performed on every sample from the sensory analysis, except for the GW11a sample for which insufficient material was available to perform the analysis. For enumeration, 10 g of mixed Gwell (collected during the first step of the sensory analysis and analyzed the next day) was diluted in 90 g citrate diluent (20 g/L trisodic citrate, pH 7.5) and homogenized in a sterile filter bag (BagPage+, Interscience, France). This initial dilution was then serially diluted in sterile diluent (tryptone 1 g, NaCl 8.5 g, H_2_O 1 L, pH 7). 100 μL of the serial dilution was plated on different nutritive and selective media (M17, MRS, PCA, VRBL, KF and OGA; see list in Table 2) using a Spiral system (Interscience, France) and incubated under aerobic and anaerobic conditions, depending on the medium (Anaerocult^®^ A, Merck, Darmstadt, Germany).

### 2.4. Bacterial Isolation and Phenotypic Sorting

The five Gwell considered as typical on completion of the sensory evaluation were selected to characterize their bacterial diversity. Strains were isolated from the M17, PCA and MRS agar plates used for enumeration, incubated at 30 °C and 43 °C under aerobiosis and anaerobiosis, respectively. Around six colonies of each morphology were isolated from M17 medium, while two or three colonies were isolated from MRS agar plates incubated at 43 °C and from PCA agar plate. The isolates were then sorted according to macroscopic criteria (planktonic growth, widespread or concentrated at the bottom of the culture tube), their ability to ferment ribose and microscopic criteria (cell morphology, bacilli or cocci, length of cocci chains). When several clones isolated from the same samples and with the same combinations of growth media and culture conditions displayed similar microscopic and macroscopic profiles and ribose fermentation profile, only one clone was selected for subsequent identification steps. An enrichment step was added to verify the presence of thermophilic LAB species and particularly thermophile *Streptococci*, which might be present at very low levels. The Gwell samples were inoculated (10% (*v*/*v*)) in thermized and skimmed milk incubated at 43 °C, and the resulting samples were then plated on M17 agar plates incubated for 48 h at 43 °C, the resulting colonies being isolated for identification.

### 2.5. Identification and Phenotypic Characterization of Strains

#### 2.5.1. API 50CH—Biochemical Characterization

All the selected isolates were subjected to phenotypic characterization in order to determine a profile of their sugar fermentation abilities using API 50CH profiles (API System, BioMerieux, Marcy l’Etoile, France). The fresh culture was centrifuged and the pellets suspended in API medium according to the manufacturer’s instructions. The Apilab Plus computer-assisted identification software (API-bioMerieux, Basingstoke, UK; version 4.0) was used to analyze the carbohydrate fermentation profiles.

#### 2.5.2. Identification of the Biovar *diacetylactis*

Strains identified as *Lactococcus lactis* by API gallery were cultured on KCA medium [21] incubated at 30 °C for 3 days. KCA medium makes it possible to distinguish colonies as a function of their ability to use extracellular citrate. A translucent halo appears around a colony when it consumes the surrounding calcium citrate and this means that the strains belongs to the *Lactococcus lactis* sp. *lactis* biovar *diacetylactis* species.

#### 2.5.3. DNA Extraction for 16S rDNA Sequencing and Species-Specific PCR

The total DNA of selected isolates was extracted using the DNeasy Blood and Tissue kit (Qiagen ref 69504; Hilden, Germany). We followed the manufacture’s protocol for Gram positive bacteria, but with two modifications: (i) the 1 mL pellet of pure culture was treated with a lysis buffer containing additional lysozyme and mutanolysin (20 mM Tris-HCl, pH 8.0, 2 mM EDTA, 1% Triton X100, 20 mg/mL lysozyme and 50 U/mL mutanolysine) at 37 °C for 1 hour; (ii) the proteinase K step was achieved at 56 °C for 30 min. DNA concentrations were quantified at a wavelength of 260 nm using a Nanodrop DN-1000 spectrophotometer (Labtech, Palaiseau, France).

#### 2.5.4. 16S rDNA Sequencing

The 16S rDNA gene of one atypical strain was amplified by W001 and W002 primers according to the method described by Godon et al. (1997) using a Thermal Cycler C1000™ (Bio-Rad, Gladesville, Australia). Amplified PCR products were sequenced using the Sanger method by LGC Genomics (Berlin, Germany). The sequences were assembled using VectorNTI software (Invitrogen, Carlsbad, United States) and submitted to the National Center for Biotechnology Information GenBank (NCBI, Bethesda MD, 20894 USA, www.ncbi.nlm.nih.gov). A basic local alignment (BLASTn) was performed to determine by homology the best identification of the isolates.

#### 2.5.5. Species-Specific PCR

In order to identify the isolates collected after phenotypic sorting and to differentiate subspecies of *Lactococcus lactis*, the specific primers used for *Lactococcus lactis* were: 5${}^{\prime }$-TTTGAGAGTTTGATCCTGG-3${}^{\prime }$ and 5${}^{\prime }$-GGGATCATCTTTGAGTGAT-3${}^{\prime }$ according to Pu et al. [22] and 5${}^{\prime }$-TTATTTGAAAGGGGCAATTGCT5514-3${}^{\prime }$ and 5${}^{\prime }$-GTGAACTTTCCACTCTCACAC-3${}^{\prime }$ for *Streptococcus thermophilus*, according to Forsman et al. [23].

#### 2.5.6. Pulse-Field Gel Electrophoresis (PFGE)

The clonal diversity of *Lactococcus lactis* was determined by PFGE analysis of the strains after phenotypic sorting. Bacterial cultures and agarose plugs were prepared, as described previously by Lortal et al. [24]. The plugs were equilibrated for one hour in a restriction buffer (CutSmart, New England Biolabs, Beverly, MA, USA) at 4 °C, and were then transferred to a fresh digestion buffer containing 15 units SmaI endonuclease (New England Biolabs, Beverly, MA, USA) for one hour. The plugs were then incubated at 25 °C for 4 h. PFGE was performed in a Bio-Rad CHEF DRII electrophoretic cell on 1% (*w*/*v*) agarose gel (Ultrapur, Gibco-BRL, Inchinnan, Scotland) in 0.5x TBE buffer (45mM Tris, 45mM boric acid, 1mM EDTA, pH 8.0) at 200 V and 14 °C, under the following conditions: initial time—2 s, final time—25 s, total running time—21 h. Strain CIRM-BIA 127 was used as a home-made ladder for this study. After migration, the gels were stained with gelRed, visualized using UV light and then analyzed with GelCompar software (BioNumerics, Applied Math, Austin, TX, USA). Conversion, normalization, and further analysis were performed using the Pearson coefficient and the Unweighted Pair Group Method with Arithmetic Mean (UPGMA) cluster analysis with BioNumerics software (Applied Math, Sint-Martens-Latem, Belgium).

#### 2.5.7. Acidifying Capacity of Strains

The strains selected after phenotypic sorting were grown overnight in M17 broth at 30 °C. They were then harvested by centrifugation for 5 min at 5000 g and washed with sterile tryptone water 1% (*w*/*v*) (HiMedia Laboratories, Einhausen, Germany). The acidifying curve of LAB was subsequently assayed in sterile UHT skimmed milk incubated at 30 °C (5% (*v*/*v*) in 20 mL) using the Cinac system (Ysebaert, Frépillon, France).A positive control was performed with Gwell samples inoculated at 30 °C (10% (*v*/*v*)) in UHT skimmed milk incubated in the same conditions. The pH was recorded every 5 min for 24 h.

### 2.6. Metabarcoding

We analyzed 13 samples in November 2018 by metabarcoding, 18 months after the sensory and microbial analysis. Of the 10 Gwell producers who participated in the first part of this study, nine sent us a sample. Only GW8 did not send a sample as he was no longer producing Gwell. At this stage, four producers joined the research program. All samples were sent by post and frozen at −20 °C on reception.

#### 2.6.1. DNA Extraction

DNA was extracted from the Gwell samples using the Nucleospin Tissue kit (Macherey-Nagel, Düren, Germany). The samples were diluted 10-fold in 20 g/L citrate solution heated at 42 °C and homogenized in a filter bag (BagPage+, Interscience); 5 mL was then centrifuged at 5000 g for 10 min. The supernatant was discarded and the pellet was resuspended in 1.5 mL citrate solution and centrifuged again. The pellet was then resuspended in 400 μL lysis buffer supplemented with 20 g/L lysozyme, 50 U/mL mutanolysine and 500 U/mL lyticase to break down the cell walls. The lysis suspension was incubated for 1 h at 37 °C and then the kit instructions were followed to extract the DNA. 50 μL elution buffer was heated to 70 °C and incubated for 5 min on the membrane before elution, which was performed twice in two different tubes. Eluted DNA was quantified in each tube using a Nanodrop ND-1000 spectrophotometer (Labtech, Palaiseau, France).

#### 2.6.2. Sequencing on Illumina MiSeq

DNA sequences were amplified in the 16S V3-V4 region for bacteria and the ITS2 region for yeasts. The primers used in this study were based on 16SV3V4 Forward-TACGGRAGGCAGCAG and 16SV3V4 Reverse-TACCAGGGTATCTAATCCT for bacteria [25] and ITS2 Forward-CTAGACTCGTCATCGATGAAGAACGCAG and ITS2 Reverse-TTCCTSCGCTTATTGATATGC for fungi [26] Primers targeting these regions were designed with an Illumina tail. In order to reduce interference during Illumina sequencing, we also added frame-shifts of 4, 6 or 8 random nucleotides for the forward primers and 4, 5 or 6 nucleotides for the reverse primers, between the target sequence and the Illumina tail. All the primers used are listed in Appendix A. For each forward or reverse primer, an equal mix of the three primers containing the different frame-shifts was added to the PCR mix. For the preparation of multiplexed Illumina libraries, we employed a strategy based on a two-step PCR approach: a first PCR using the locus-specific primers including the Illumina adapter overhang (with 30 cycles), and a second PCR for the incorporation of Illumina dual-indexed adapters (with 12 cycles). Bead purifications were carried out after each PCR step. Quantification, normalization and pooling were carried out before sequencing on Illumina MiSeq [27].

#### 2.6.3. Bioinformatics Analysis

Reads were merged using Pear software v. 0.9.11, adapters removed with cutadapt v. 1.12 and Python 2.7.13 and the reads were trimmed with Sickle v. 1.33 with a quality threshold of 20 on a window of 20 nucleotides. The sequences were then analyzed using the frogs pipeline [28], the Frogs preprocess tool v. r3.0-3.0 being used to dereplicate reads, while clustering was performed using frogs clustering v. 3.0-1.4 with a distance parameter of 3. Chimera were removed with frogs chimera v. 3.0-7.0 and Operational Taxonomic Units (OTU) were filtered on and abundance superior to 0.00005 total abundance. For ITS reads, conserved ITS regions were removed using ITSX v. 1.0.11 and affiliations were assigned to OTUs with frogs affiliation v. 3.0-2.0, using the Unite Fungi database v. 8.0 [29] for ITS and Silva 132 [30] for 16*S*. Multi-affiliations were partially resolved using frogs affiliation post-process tool v. 3.0-1.0 with identity and coverage parameters of 99.5%. Remaining multi-affiliations were treated by assigning the lowest common taxonomy level to multi-affiliated OTUs. The samples were rarefied to the minimum number of reads for each barcode (89,239 for 16S and 137,468 for ITS2), using the rarefy_even_depth function of the R (v. 3.6.3) phyloseq package v. 1.24.2. The random seed was set at 5907.

### 2.7. A Participatory Research Framework

This study was initiated by the Gwell producers. As well as performing the sensory analysis and submitting their samples and data, they participated in developing the research protocol throughout the study. Meetings with all the producers involved in the study were organized after each step in order to present the results and organize the next analysis. A monitoring committee comprising five Gwell producers followed the study more closely through frequent telephone discussions.

## 3. Results

### 3.1. Consensual Descriptors of Gwell Samples Considered to Be Typical

The first step in the sensory analysis consisted in determining from the tested samples those which could be considered as typical Gwell. This was performed by an expert jury whose 15 members knew the product well and consumed it regularly as most of them were Gwell producers. The panelists could be considered to be reliable because they gave the same classification to the duplicated samples on average in 62.5% of cases. The maximum of concordance was 100% for one panelist who gave the same classification to all duplicated samples, while the least reliable panelist gave the same classification to 30% of the duplicates. Four samples (GW6a, GW5a, GW9a and GW1a) were deemed to be typical Gwell by more than 50% of the panelists (53% to 63%). Five samples (GW2a, GW3a, GW4a, GW7a and GW11a) were deemed to be typical Gwell by between 30% to 47% of the panelists, and could not be considered as typical Gwell. One sample (GW8a) was deemed to be typical Gwell by 3.3% of the panelists, and was thus clearly considered as non-typical (Figure 1).

The associated descriptors generated are summarized in a wordcloud (Figure 2). These descriptors were standardized in 38 different categories so as to enable the statistical analysis. The Gwell judged to be typical were described as having a “tangy taste” by eight panelists, and a “dense” and “smooth” texture by seven and six panelists, respectively, while the non-typical Gwell had a “yogurt taste” and were considered to be “too acidic” by seven and five panelists, respectively.

The second step of the sensory analysis consisted in a sorting task associated with a verbalization task, performed by the jury with 29 members who did not know the product. This revealed a distribution of eight samples in two distinctive groups and one isolated sample, as illustrated by the MCA analysis (Figure 3). The two MCA dimensions retained 41.2% of variance, with dimensions 1 and 2 accounting for 20.8% and 20.4% of variance, respectively. More than half of the panelists (57%) associated the duplicated samples in the same group so they were therefore very close on the MCA graph. Dimension 1 discriminated Gwell considered to be typical and non-typical, whereas dimension 2 discriminated the GW4 sample. The larger group comprised five samples (GW1a, GW5a, GW6a, GW7a and GW9a), four of which were considered to be typical according to the first sensory analysis. The second group comprised three samples (GW3a.1, GW3a.2 and GW8a) and contained both duplicated samples of GW3 and the Gwell considered as non-typical (GW8a). The isolated sample (GW4a) considered to be mildly typical during the first sensory analysis, was judged as being different from all other Gwell samples by 14 of the 28 panelists, and was thus clearly separated from both other groups.

### 3.2. Microbial Enumeration of Gwell

The enumeration results regarding nine Gwell on seven different media are presented in Figure 4. Lactic mesophilic bacteria enumerated on M17 and MRS media incubated at 30 °C largely dominated the ecosystem with levels of 8 to 9 log CFU/mL (Colonies Forming Unit per mL) Yeasts enumerated on OGA were found in four out of ten Gwell (GW4a, GW5a, GW7a and GW10a) at levels between 4 and 6 log CFU/mL, whereas molds also enumerated on OGA were found in every Gwell tested but at lower levels (2 to 4.11 log CFU/mL). Presumptive thermophilic LAB (enumerated on MRS medium incubated at 43 °C under aerobiosis) were detected in seven Gwell samples (GW1a, GW2a, GW3a, GW6a, GW7a, GW8a and GW9a). Presumptive thermophilic *Streptococci* (enumerated on M17 medium incubated at 43 °C under aerobiosis) were detected in three Gwell samples (GW3a, GW2a and GW8a), which were considered to be mildly or non-typical during the first step of sensory evaluation. Gwell samples GW7a and GW2a displayed small quantities of coliforms, with 1.8 and 2.8 log CFU/mL respectively. Yeasts were never detected with thermophilic bacteria grown on MRS at 43 °C; only the GW7a sample contained both types of microorganisms. None of the Gwell considered as typical contained thermophilic *Streptococci*; either thermophilic lactic acid bacteria or yeasts were found, but never both. Gwell considered as non-typical could contain either thermophilic *Streptococci* or coliforms. The GW8a sample, which was unanimously considered as non-typical, displayed a different microbiological profile. This sample contained 8.9 times more *Lactococci*, and 50 times more thermophilic *Streptococci* and molds compared to the mean of other Gwell samples.

### 3.3. Selection of Five Gwell Samples for the Culture-Dependent Characterization of Lab Diversity

By combining the sensory and enumeration results, five Gwell samples, GW1a, GW4a, GW5a, GW6a and GW9a, were selected for the characterization of LAB diversity using culture-dependent methods. These five samples shared the following characteristics: they were judged as typical at a threshold superior or close to 50% and levels of mesophilic LAB ranging from 8.1–8.9 log CFU/mL and possibly also yeasts at a level of around 5 log CFU/mL, but no streptococci or coliforms.

### 3.4. Characterization of LAB Diversity

#### 3.4.1. Selection and Identification of Isolates Representative of LAB Diversity

For the five previously selected Gwell samples, 100 clones were picked from the M17 and MRS agar plates. In order to eliminate duplicated clones, they were subjected to a sorting task according to their macroscopic and microscopic characteristics as described in Materials and Methods. They were discriminated in terms of their colonies, planctonic growth and cell morphology. Strains isolated from the same medium and having similar macroscopic and microscopic characteristics were considered to be identical; 26 different strains were thus selected as a result of sorting (Table 3).

Following the enrichment step, it was found that 23 isolates came from cultures under mesophilic conditions and three from thermophilic conditions after the enrichment step. Among the 23 mesophilic strains, 22 were identified as a function of their metabolic sugar fermentation abilities using API 50CH profiles. Ten strains were identified as *Lactococcus lactis* sp. *lactis*, 12 as *Lactococcus lactis* sp. *cremoris* (with a total for both subspecies of 13 different API profiles). One strain was identified as *Staphylococcus warnerii* by rDNA 16S sequencing. All the *Lactococcus* strains also underwent species-specific PCR in order to confirm the identification results. The three thermophilic strains were identified as *Streptococcus thermophilus* using species-specific PCR. All the results are summarized in Table 3. The culture of *Lactococcus* strains on KCA medium revealed that all the *L. lactis* sp. *lactis* isolated were citrate positive, which means that all *L. lactis* sp. *lactis* belonged to the biovar *diacetylactis*. PFGE profiles were determined for the 22 *L. lactis* strains in order to evaluate intra-species genetic diversity. Fifteen different patterns were obtained and revealed a high level of intra-species diversity for both subspecies. Nine different PFGE patterns were identified for the ten *L. lactis* sp. *lactis* biovar *diacetylactis* strains and six for the 12 *L. lactis* sp. *cremoris* strains. Isolates displaying the same API & PFGE profile and isolated from the same Gwell sample were considered to be a single strain. This was the case for isolates Q&C (GW1a samples) and I&G (GW5a samples). Thus among the 22 *L. lactis* isolates sorted, 20 could be considered as single strains. The same PFGE profile was found in two different Gwell samples (GW6a and GW9a) associated with the K and P strains, respectively, and they also shared the same API pattern, so both isolates very probably corresponded to the same strain. We also noted that the same PFGE pattern could correspond to different API patterns; for example, strains D and O shared the PFGE pattern P15 but varied in terms of three different substrates on the API pattern, so those isolates could therefore be considered as different strains with different functionalities. Among the five Gwell samples analyzed, GW5a displayed the highest level of diversity in terms of the number of strains (six different strains) associated with four different API patterns and five different PFGE patterns.

#### 3.4.2. Acidifying Capacity of *Lactococcus*
*lactis* Strains

The milk acidifying capacity was measured for all *Lactococci* strains (Figure 5). Two groups could easily be distinguished: one fast-acidifier group, containing only sup-species *cremoris* strains, and a slow-acidifier group containing all subspecies *lactis* biovar *diacetylactis* strains and four *cremoris* strains with an intermediary acidification phenotype (O, U, H and K strains). The mean acidification rate between 0 and 5 h was 0.18 (standard deviation = 0.069) u.pH/h for *cremoris* strains and 0.08 (sd = 0.0078) u.pH/h for *lactis* strains, and the mean final pH after 24 h was 4.44 (sd = 0.48) for *cremoris* strains and 5.18 (sd = 0.080) for *lactis* strains. *S. thermophilus* strains displayed similar acidifying capacities for the three isolated strains, with an acidifying rate that was intermediate between *lactis* and *cremoris* strains at 0.10 (sd = 0.016) u.pH/h.

### 3.5. Culture-Independent Characterization of Gwell Microbial Communities

The 13 Gwell samples collected in the second test batch were analyzed using the metabarcoding approach in order to obtain an overview of the bacterial diversity associated with Gwell and to supplement our results obtained using a culture-dependent approach. Each producer whose Gwell was analyzed using the culture-dependent method submitted a sample, except for GW8 who had meanwhile stopped producing Gwell. The results are presented in Figure 6B,C. We obtained between 89,239 and 113,555 reads for 16S and between 137,468 and 180,774 reads for ITS2 sequencing. The mean Shannon indices were 1.04 (sd = 0.57) and 0.15 (sd = 0.22) with 16S and ITS2 sequencing, respectively. After filtering only Operational Taxonomic Units (OTUs) accounting for more than 0.5% of sample abundance, thus retaining more than 96% of reads for bacteria and 99.5% of reads for fungi, the mean numbers of species per sample were 4.53 and 1.40 for bacteria and fungi, respectively. *Lactococcus lactis* was found in every sample and it was the most abundant species. Three OTUs belonging to the genus *Lactococcus* could not be identified at the genus level. Two *Streptococcus* species were identified in eight samples: *S.* saliviarius and *S. saliviloxodontae*. The *Streptococcus salivarius* OTU corresponds to the subspecies *thermophilus*, which is synonymous with *Streptococcus thermophilus* (Farrow and Collins, 1984). *Lactobacillus helveticus* was detected in two samples and *Lactobacillus acidophilus* in one. *Geotrichum candidum* was the only filamentous fungal species to be identified; it was found in all samples and was by far the most abundant fungal species. Concerning yeasts, *Kazachstania servazii* was detected in three samples (GW4, GW5 and GW10) and *Yarrowia lipolytica* in only one sample (GW7b). As for intraspecific diversity, we found a total of eight *Lactococcus* OTU and five *Streptococcus* OTU. The mean numbers of *Lactococcus* and *Streptococcus* OTU per sample were both 3.51 (Appendix A). The most abundant *L. lactis* OTU was found in every sample. The five less abundant *Lactococcus* OTUs were found in one to five different samples. Interestingly, two other *Lactococcus* OTUs were only found in the eight Gwell (GW1b, GW2b, GW3b, GW6b, GW9b, GW12b, GW13b and GW14b) that contained *Streptococcus*. We detected one *G. candidum* and one *K. servazii* OTU across all samples.

### 3.6. The History of Gwell Exchanges between Producers Explains Their Microbial Composition

When they start a new production or have lost their Gwell ferment, producers obtain their ferment from one of their colleagues. By integrating the history of ferment exchange data between producers, we were able to draw a genealogy for Gwell (see Figure 6A). Two clusters could be distinguished based on Gwell exchanges and the times of in-farm differentiation. Cluster 1 grouped nine Gwell whose common ancestor was a Gwell from producer GW9. Cluster 2 comprised four Gwell that all derived from GW16 production. All samples containing *S. thermophilus* were in cluster 1, and only GW7b (which had diverged from GW9b 17 months before analysis) did not have it. The GW5b and GW11b samples in cluster 2 were the only ones to contain *Lactobacillus helveticus*, whose abundance was higher in GW11b than in its parent GW5b. The GW4b, GW5b and GW10b samples all contained the same *K. servazii* OTU, which was consistent with the common origin of these Gwell. This species was not detected in GW11b even though it had only diverged from the GW5b Gwell ten days prior to the analysis, indicating that it had undergone a maximum of four back-slopping processes since the GW11 producer received his ferment from GW5.

## 4. Discussion

The first step in this study consisted in defining a typical Gwell and describing it from an organoleptic point of view, so that its microbial composition could then be analyzed. Because the product had not previously been precisely characterized and production processes could still be quite diverse, we wanted to ensure that we studied samples that were representative of what the product should be. The study therefore enabled farmers to define consensual descriptors of their product, which constituted an important step in their collective efforts to revive this traditional fermented milk, while at the same time offering us an opportunity to document its microbial composition and the effects of exchanges between farmers.

### 4.1. Microbiological Characterization of Gwell

The analysis of microbial composition permitted by cultures on selective media revealed that *L. lactis* and a fungus with the morphology of *G. candidum* were found in all the samples. Studies performed by a private laboratory had already identified both *L. lactis* and *S. thermophilus* in Gwell [31]. Species diversity was low in all samples. Indeed, even with stringent filters retaining only the most abundant OTU and the five most abundant species, we recovered almost all reads in every sample (more than 99.5% for fungi and more than 96% for bacteria). It is relevant to apply the concepts developed in ecology and community assembly to understand this observation. Gänzle et al. showed for another fermented product, sourdough, that selection was the principal evolving force that determines microbial composition [32], thus explaining the low level of species diversity by competitive exclusion. As Gwell is produced using a similar process of back-slopping, we can suppose that the same explanation applies. Moreover, dispersal is presumably even lower insofar as Gwell is produced using pasteurized milk. This means that the only source of microorganisms inoculated in addition to the ferment is the processing environment via the equipment used and the atmosphere. Although at relatively low levels, *G. candidum* was found in all the samples analyzed, using both culture-dependent and culture-independent approaches. Of the 13 producers who participated in the study, 12 produced cheese containing *G. candidum* in the same room as that used for Gwell (data missing for one producer). This mold probably arises from cross-contamination between production runs and may be stably selected at a low level throughout successive back-slopping procedures, or always reinoculated into Gwell from the production environment. The five Gwell samples considered to be typical were analyzed further using a culture approach. Their ecosystem was largely dominated by *Lactococcus lactis* with both subspecies *lactis* biovar *diacetylactis* and *cremoris*. This species is found in other back-slopped mesophilic fermented milks throughout the world. For example, it has been detected in Viili, which is very similar to Gwell in terms of the technologies employed, involving back-slopping inoculation and mesophilic fermentation. Its microbial composition is very close to that of Gwell, with strains of *L. lactis* sp. *cremoris*, *L. lactis* sp. *lactis* and *G. candidum* dominating its microbial ecosystem. The main difference is that all subspecies *lactis* belong to the biovar *diacetylactis* in Gwell, whereas some strains were identified as subspecies *lactis* but not biovar *diacetylactis* in Viili [13]. Both subspecies of *L. lactis*, *lactis* and *cremoris*, have also been identified in back-slopped goat cheese and in goat’s milk from different regions of France where they represent the dominant bacterial population [5]. In their study, the authors found a predominance of *lactis* subspecies strains which accounted for 80% of the *L. lactis* strains isolated when compared to *cremoris*. The L. lactis species has also been identified as the principal LAB species in the South African and Indian fermented milks aMasi, Dahi and Dadhi, whose technologies are very similar to Gwell [16,33,34]. The two subspecies were distinguished using metabolic and molecular tests and displayed different characteristics. *L. lactis* sp. *lactis* biovar *diacetylactis* produced volatile compounds that are generally associated with aroma and flavors [35]. Using GC-Mass Spectrometry analysis, we confirmed that the subspecies *lactis* biovar diacetylatics produced diacetyl and aroma-related compounds whereas subspecies *cremoris* did not; these aroma-related compounds were also found in all the Gwell samples analyzed (data not shown). The presence of *S. thermophilus* in Gwell samples was revealed by both metabarcoding analysis and culture-dependent analysis in some samples; this was quite surprising since the mesophilic technology used for this product should disadvantage the growth of thermophilic LAB. However, it could be explained by cross-contamination from yogurt or cheese production, which is performed in parallel by most of the Gwell producers.

### 4.2. Microbial Composition of Typical and Non-Typical Gwell

Gwell judged as typical never contained coliforms or thermophilic streptococci. Moreover, Gwell deemed to be non-typical were described as having a yogurt taste. *S. thermophilus* is a LAB species that is used to produce yogurt [36] and it was identified in typical Gwell samples following an enrichment step. We can hypothesize that the thermophilic streptococci identified in non-typical Gwell were *S. thermophilus* and were present at excessively high levels, thus giving it a yogurt taste. *Streptococcus* species were however identified by metabarcoding on 16S V3V4 rDNA in eight out of 13 samples, sometimes with a greater abundance than that of *L. lactis*. The differences observed in the composition of microbial species may probably only partly explain the sensory differences seen between typical and non-typical products. The production process is an important factor determining the organoleptic properties of fermented milks [37]. For example, the fermentation temperature may determine the quantity and quality of exopolysaccharides [38] or aroma [39] produced by LAB, thereby influencing the texture and taste of Gwell. Variations in the fermentation temperature occur between producers as they use different systems to maintain temperature during fermentation. Some use professional grade or homemade ovens with temperature control, while others use isothermal recipients that only maintain the initial temperature. This factor could be more important than differences in the microbial species composition of Gwell to explain their organoleptic differences.

### 4.3. The Evolution of Gwell over Time and Exchanges between Producers

We were able to identify two Gwell clusters based on Gwell exchanges where the microbial composition was markedly homogeneous in each cluster. In cluster 1, *Streptococcus* species were found in all but one sample and yeasts in only one, whereas in cluster 2, yeasts were identified in all but one sample and no *Streptococcus* species were found. The main *L. lactis* OTU was shared among all samples, but cluster 1 had two specific OTUs that were not found in cluster 2. These results were consolidated by PFGE analysis of the isolated strains, and indeed the PFGE profile P14 was found in GW1a, GW6a and GW9a, which all belonged to cluster 1 while it was not found in GW4a and GW5a which were part of cluster 2. The microbial composition of Gwell is thus highly conserved throughout Gwell exchanges and over time. *S. thermophilus* was also maintained throughout the Gwell exchanges. It was detected in all the samples in cluster 1, the only exception being GW7b, which diverged for 17 months from GW9b. It did not contain *S. thermophilus* but a yeast was detected, so a different environment may have modulated the microbial community. Yeasts were found in both typical and non-typical Gwell. The three Gwell (GW4a, GW5a and GW7a) in which yeasts were detected using culture-dependent methods in the first analysis were continuously back-slopped by the producers until collection of the sample for the culture-independent analysis 18 months later. Yeasts were still found in these samples at an abundance ranging from 8000 to 38,000 reads, for a total of 137,000 reads per sample. They were identified as the same OTU of *K. servazii*, suggesting that this yeast stably colonized the ecosystem. This species had already been identified in various fermented foods, including beer [4], or sourdough [40,41]. The *Kazachstania* genus is often detected in the sourdough used for bread production [40] and can also be found at low levels in grape must where it may be involved in the formation of wine aroma [42]. More generally, however, it has been associated with spoilage in dairy fermented products and more specifically cheese [43] although *K. servazii* was nevertheless found at a significant level in camembert-like cheese by Mei et al. [44]. The presence of yeast was probably specific to the producer’s practices as *K. servazii* were detected in the GW5b sample but are not recovered in that GW11b sample that had come from the GW5 sample back-slopped for 10 days (fewer than four successive back-slopping processes). These yeasts had probably been counter-selected by the technological process used by GW11.

### 4.4. Gwell Loss

This study may help us to understand the dynamics of Gwell in the context of regular back-slopping over a long period of time. Indeed, producers sometimes experience slowed-down acidification, leading to the loss of their ferment as it cannot be used safely to inoculate the next production run. According to our results, this reduction in acidification rate could be explained by a drop in the number of *cremoris* subspecies strains, as they acidify milk faster than *lactis* strains. Several hypotheses might explain a decrease in a specific strain population, for example a phage attack. Phage attacks have been reported during the production of many fermented milks [45]. Community susceptibility to phages is dependent on the diversity of their host resistance mechanisms [46]. We were unable to assess this diversity, but we observed between two and five different PFGE profiles of *Lactococcus lactis cremoris* in the same Gwell. Different *Lactococcus* OTUs could be identified by metabarcoding on the 16S rDNA V3V4 region, even though this region is highly conserved. These observations suggest that different *Lactococcus lactis* lineages coexist in a Gwell sample. If different *L. lactis* sp. *cremoris* strains with different resistance mechanisms coexist, then phages could not be a threat to the Gwell ecosystem. Another hypothesis concerns the dynamic balance between the two subspecies, with sp. *lactis* out-competing sp. *cremoris* and causing a reduced acidification rate. A loss of Gwell could also be explained by a loss of sp. *lactis*, or even *G. candidum* if a mechanism of nutritional exchanges exists between them and sp. *cremoris*, as has been demonstrated between S. *thermophilus* and L. *delbrueckii* in yogurt [47]. Abiotic factors controlled by the production process, such as the fermentation or conservation temperatures or the frequency of back-slopping, and biotic factors such as the inoculation rate, may also affect the respective proportions and activity of the different species in a Gwell microbial community and thus explain its evolution over time. Interactions between sp. *lactis*, sp. *cremoris*, G. *candidum* and their environment need to be studied in order to understand how and whether they can coexist over the long-term in the Gwell ecosystem.

## 5. Conclusions

During this work, participatory sensory analysis was able to produce a consensual definition of the organoleptic properties of a typical Gwell. We identified *L. lactis* as the main species in the Gwell ecosystem, with both subspecies systematically being associated: *L. lactis* sp. *cremoris* and *L. lactis* sp. *lactis* biovar *diacetylactis*. *G. candidum* was also identified in all the samples. Using complementary metabarcoding experiments, we identified two separate Gwell lineages by reconstructing the history of Gwell exchanges. We showed that each lineage had a different microbial composition, with the presence of *K. servazii* and *S. thermophilus* depending on the lineage considered. Describing the microbial community in Gwell enabled us to formulate the hypothesis that episodic losses of the product are caused by an imbalance between *L. lactis* sp. *cremoris* and *L. lactis* sp. *lactis*, and possibly *G. candidum*. The next stage in this study will therefore consist in identifying factors that influence the development of both *L. lactis* subspecies in the Gwell ecosystem.

## Figures and Tables

**Figure 1 microorganisms-08-00982-f001:**
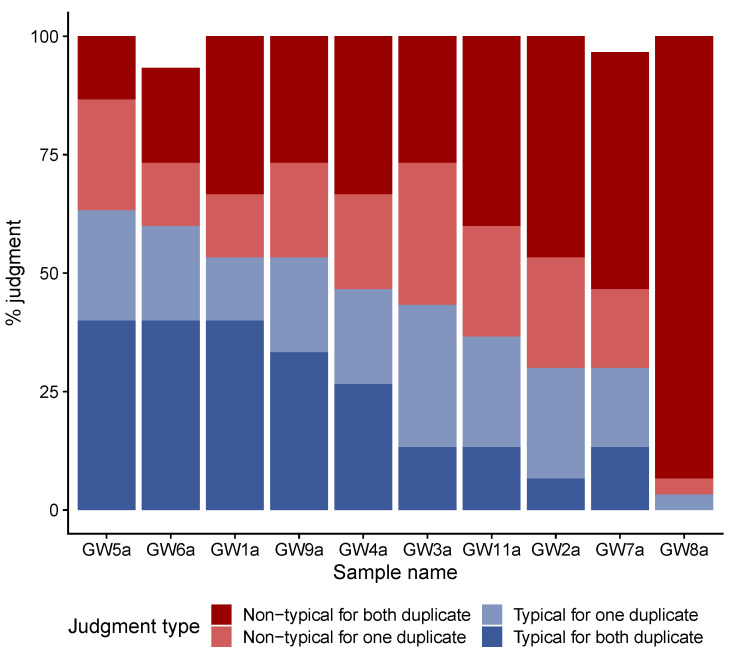
Results of the first step of the sensory analysis performed with the expert panel of 15 judges. Bar height represents the percentage of judgements for each category. Dark blue bars represent typical judgments for both duplicate of a sample, light-blue bars typical judgements for one of the two duplicate of a sample, light-red bars non-typical judgements for one of the two duplicate of a sample and dark-red bars non-typical judgements for both duplicate of a sample.

**Figure 2 microorganisms-08-00982-f002:**
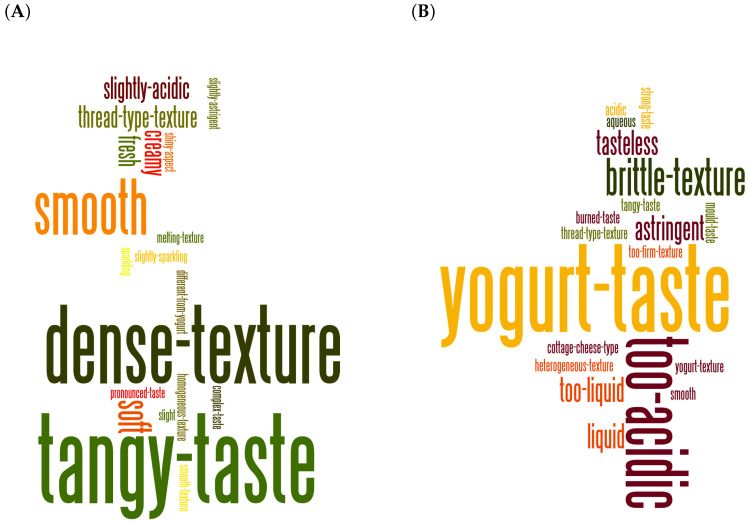
Wordcloud of descriptors used for typical and non-typical Gwell. Word sizes are proportional to the number of citations. (**A**), Gwell considered as typical by panelists; (**B**), Gwell considered as non-typical by panelists.

**Figure 3 microorganisms-08-00982-f003:**
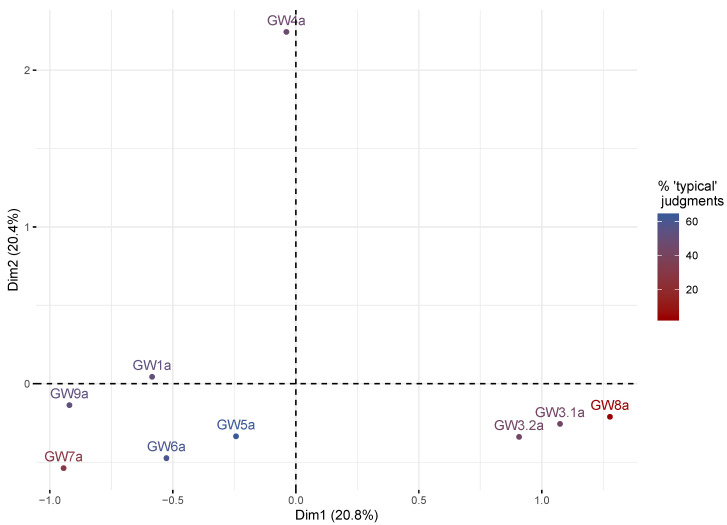
Multiple Correspondence Analysis of the Gwell sorting task performed by the non-expert jury. Each sample is colored according to the percentage of times it had been considered to be typical during the previous analysis.

**Figure 4 microorganisms-08-00982-f004:**
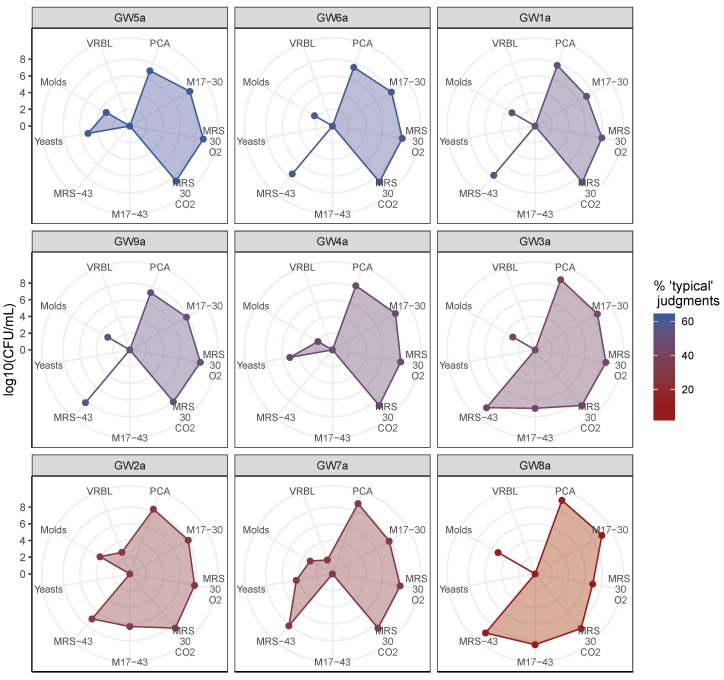
Radar graph of microbial counts. Each radar axis shows the enumeration results expressed in log CFU/mL for one medium. The name of each medium is followed if necessary by the incubation temperature (30 °C or 43 °C), and the aerobic (O_2_) or anaerobic (CO_2_) conditions applied. Yeasts and molds were cultured on OGA medium. All radar axes had the same scale. Samples are ordered and colored according to the typicality score obtained during the first sensory analysis.

**Figure 5 microorganisms-08-00982-f005:**
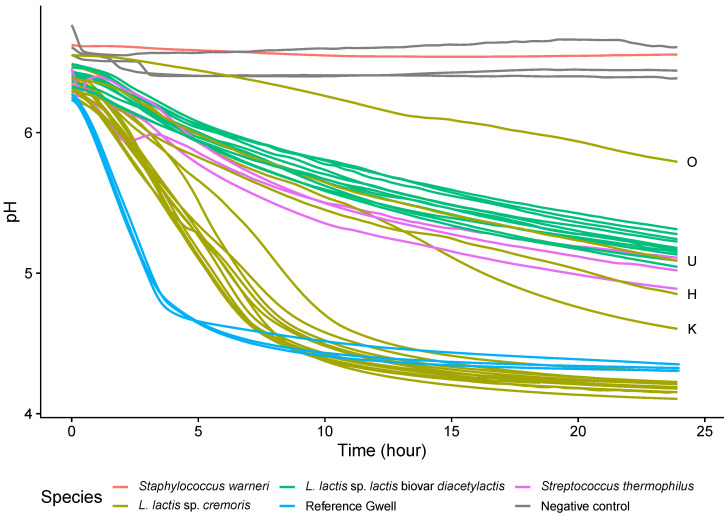
Acidification curves for the 26 isolated and selected strains inoculated (5% (*v*/*v*)) in sterilized skimmed milk incubated at 30 °C for 24 h. The pH curves are colored according to the species. Acidification curves of reference Gwell are shown in blue. Atypical slow-acidifier *L. lactis* sp. *cremoris* strains are labeled with their clone ID letter.

**Figure 6 microorganisms-08-00982-f006:**
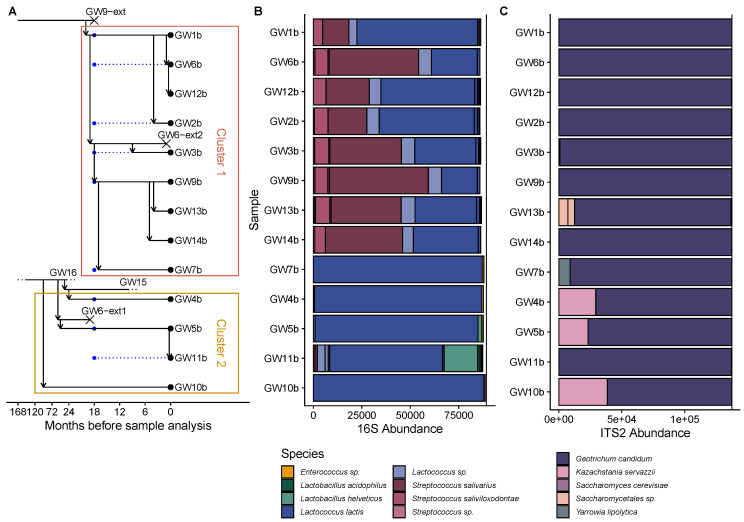
History of ferment exchanges and species identified in Gwell using 16S and ITS2 metabarcoding. (**A**) Gwell exchange diagram. The horizontal axis represents the time in months. The scale has a 6-month step between 0 and 24 months, and a 48-month step between 24 and 168 months. Horizontal lines represent the period of time during which each producer continuously back-slopped his Gwell. A vertical arrow pointing from producer X to producer Y represents the acquisition of a ferment by producer Y from producer X.. Samples that were analyzed during this study are symbolized by a point. Blue points represent Gwell that were analyzed using culture-dependent methods and black points represent samples that were analyzed with culture-independent methods. Gwell that became extinct are marked with a cross and labeled “ext” after the producer’s name. (**B**,**C**) Abundance of species identified by 16S V3V4 and ITS2 regions, respectively. All Operational Taxonomic Units (OTUs) whose abundance was lower than 0.5% of total sample abundance were discarded, and only the five most abundant species are shown. Species are differentiated by color and black lines inside the bars delimit different OTU. The vertical line represents the total rarefied sample abundance.

**Table 1 microorganisms-08-00982-t001:** List of the 23 Gwell samples studied in this study and respective analysis performed on it. Gwell samples share the same ID when coming from the same producers; a, corresponding to the samples analyzed in the first step of the study; b, corresponding to the samples analyzed during the second step of the study, 18 months later.

	First Gwell Sample Collection (May, 2017)					Second Gwell Sample collection (November, 2018)	
**Producer**	**Gwell Sample Code**	**Sensory Analysis**		**Microbial Enumeration**	**Bacterial Isolation**	**Gwell Sample Code**	**Metabarcoding**
		**Step1**	**Step2**				
1	GW1a	x	x	x	x	GW1b	x
2	GW2a	x	-	x	-	GW2b	x
3	GW3a	x	x	x	-	GW3b	x
4	GW4a	x	x	x	x	GW4b	x
5	GW5a	x	x	x	x	GW5b	x
6	GW6a	x	x	x	x	GW6b	x
7	GW7a	x	x	x	-	GW7b	x
8	Gw8a	x	x	x	-	GW8b	-
9	GW9a	x	x	x	x	GW9b	x
10						GW10b	x
11	GW11a	x	-	-	-	GW11b	x
12						GW12b	x
13						GW13b	x
14						GW14b	x

**Table 2 microorganisms-08-00982-t002:** List of the different media and incubation conditions used for the numeration step as well as the isolation of strains.

Name of the Medium	Composition or Reference of the Medium	Targeted Microorganisms	Temperature of Incubation	Condition of Incubation (Aerobiosis or Anaerobiosis)
M17	Biokar Diagnostic	Mesophilic and thermophilic Lactococcocal and streptococcal strains	30 °C or 43 °C	$O_{2}$
MRS	Biokar Diagnostic	Mesophilic and thermophilic lactobacilli strains	30 °C or 43 °C	$O_{2}$ & $CO_{2}$
VRBL	Difco Laboratories, detroit, MI, USA	Coliforms	30°C	$O_{2}$
KF	Biokar Diagnostic	Enterococcus	30 °C	$O_{2}$
PCA	Biokar Diagnostic	Mesophilic and aerobic total flora	30 °C	$O_{2}$
OGA	Biokar Diagnostic	Yeast and filamentous fungi	25 °C	$O_{2}$

**Table 3 microorganisms-08-00982-t003:** Identification results obtained for the 26 strains isolated from the typical Gwell samples.

Gwell Sample	Clone ID	API Strips 50CH Profile	Species-Specific PCR Identification	KCA Positive Strains	PFGE Profil Type	Validated Identification
GW1	B	-	-	-	-	*Staphylococcus warneri* (Identified by 16S sequencing)
	Q	2	*L. lactis* sp. *cremoris*	No	P13	*Lactococcus lactis* sp. *cremoris*
	C	2	*L. lactis* sp. *cremoris*	No	P13	*Lactococcus lactis* sp. *cremoris*
	R	13	*L. lactis* sp. *lactis*	Yes	P10	*Lactococcus lactis* sp. *lactis* biovar *diacetylactis*
	D	4	*L. lactis* sp. *cremoris*	No	P15	*Lactococcus lactis* sp. *cremoris*
	A	12	*L. lactis* sp. *lactis*	Yes	P6	*Lactococcus lactis* sp. *lactis* biovar *diacetylactis*
	GW1	-	*S. thermophilus*	-	-	*Streptococcus thermophilus*
GW4	F	9	*L. lactis* sp. *lactis*	Yes	P2	*Lactococcus lactis* sp. *lactis* biovar *diacetylactis*
	S	11	*L. lactis* sp. *lactis*	Yes	P3	*Lactococcus lactis* sp. *lactis* biovar *diacetylactis*
	E	5	*L. lactis* sp. *cremoris*	No	P14	*Lactococcus lactis* sp. *cremoris*
GW5	H	6	*L. lactis* sp. *cremoris*	No	P11	*Lactococcus lactis* sp. *cremoris*
	I	4	*L. lactis* sp. *cremoris*	No	P1	*Lactococcus lactis* sp. *cremoris*
	U	6	*L. lactis* sp. *cremoris*	No	P12	*Lactococcus lactis* sp. *cremoris*
	Jp	9	*L. lactis* sp. *lactis*	Yes	P4	*Lactococcus lactis* sp. *lactis* biovar *diacetylactis*
	Jg	9	*L. lactis* sp. *lactis*	Yes	P5	*Lactococcus lactis* sp. *lactis* biovar *diacetylactis*
	V	11	*L. lactis* sp. *lactis*	Yes	P4	*Lactococcus lactis* sp. *lactis* biovar *diacetylactis*
	G	4	*L. lactis* sp. *cremoris*	No	P1	*Lactococcus lactis* sp. *cremoris*
GW6	L	8	*L. lactis* sp. *lactis*	Yes	P7	*Lactococcus lactis* sp. *lactis* biovar *diacetylactis*
	M	7	*L. lactis* sp. *cremoris*	No	P13	*Lactococcus lactis* sp. *cremoris*
	K	3	*L. lactis* sp. *cremoris*	No	P13	*Lactococcus lactis* sp. *cremoris*
	GW6	-	*S. thermophilus*	-	-	*Streptococcus thermophilus*
GW9	W	10	*L. lactis* sp. *lactis*	Yes	P9	*Lactococcus lactis* sp. *lactis* biovar *diacetylactis*
	P	3	*L. lactis* sp. *cremoris*	No	P13	*Lactococcus lactis* sp. *cremoris*
	N	9	*L. lactis* sp. *lactis*	Yes	P8	*Lactococcus lactis* sp. *lactis* biovar *diacetylactis*
	O	1	*L. lactis* sp. *cremoris*	No	P15	*Lactococcus lactis* sp. *cremoris*
	GW9	-	*S. thermophilus*	-	-	*Streptococcus thermophilus*

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
