# Peer review of "Microbial Diversity Associated with Gwell, a Traditional French Mesophilic Fermented Milk Inoculated with a Natural Starter"

_microorganisms, 2020, doi:10.3390/microorganisms8070982_

Round 1
Reviewer 1 Report
Very interesting work entitled 'Microbial diversity associated with Gwell, a
traditional French mesophilic fermented milk inoculated with a natural starter'.
I figure out that Multiple correspondence analysis is like a PCA analysis so it has statistic evaluation being carried out.
I would expect Figure 5 on acidification curves to have error bars for statistical analysis.
Author Response
Response to Reviewer 1 Comments
First of all, the authors would like to thank Reviewer 1 for the time dedicated to our manuscript and his constructive comments.
Very interesting work entitled 'Microbial diversity associated with Gwell, a
traditional French mesophilic fermented milk inoculated with a natural starter'.
I figure out that Multiple correspondence analysis is like a PCA analysis so it has statistic evaluation being carried out.
> You are right; Multiple Correspondence Analysis (MCA) is effectively very close to Principal Component Analysis. We use MCA since we have to compare textual descriptors rather than numerical values. In MCA, textual descriptors are converted into numerical values and then analyzed by a PCA analysis.
More precisely, Multiple Correspondence Analysis can be seen as a PCA analysis performed on a set of categorical variables associated to the individuals. Each categorical variable is first divided into each value it can take. For each value of each variable, dummy variables are made by assigning to each individual 1 if the value was associated to it and 0 otherwise. Each dummy variable is then weighted according to the proportion of individuals possessing the corresponding value. A PCA is finally performed on this table (called the transformed complete disjunctive table).
I would expect Figure 5 on acidification curves to have error bars for statistical analysis.
> We can’t give error bars on the Figure 5 since we do not performed replicate of these kinetics for individual strains, indeed we were interested in analyzing the mean acidification rate and its standard deviation for each considered species. These data are discussed in the text. We decided however to plot the individual curves for each strain rather than the mean acidification rate of each species or subspecies with standard deviation. It allowed us to show more information while still rendering a clear and readable figure.

Reviewer 2 Report
General comments:
The manuscript titled “Microbial diversity associated with Gwell, a traditional French mesophilic fermented milk inoculated with a natural starter” has been focused on the relation between the Gwell organoleptic characteristics and the microbial community. The manuscript is interesting, it opened up a niche product (gwell fermented milk), any way it results very useful in the experimental design structure that can be applied to other types of fermented foods. The introduction is well written, the results and discussion is too confused, it is probably due to the English language.
Concluding, I suggest the manuscript publication after minor revision and an in-depth English mother tongue revision.
Suggested corrections
My first doubt concerns the “Materials and Methods” section that it’s arranged at the end of the manuscript, why? Is there a reason, or is it a mistake?
Line 377: please insert the bibliographic reference (i.e. R Core Team. R: A Language and Environment for Statistical Computing. Available online: https: //www.r-project.org (accessed on…dd mm year).
Line 385: again, you have to insert the software references.
Line 303: authors have to specify the meaning of acronym ETS.
Line 307: please wright well the reagent name i.e. Anaerocult® A. Anyone could be able to reproduce your methods in the lab.
Line 410: is the “APILAB Plus computer software (API-bioMerieux, Basingstoke, UK)”, please reports all the references.
Line 502: please insert also the reference for Illumina MiSeq
Paragraph 4.7: Authors must delete this paragraph. It is away off subject; you have inserted these considerations in the acknowledgement, it’s enough
Lines 89-94: the sentence is not clear, and the authors must rewrite it.
Figure 1: The data have not scientific value due to the absence of error bars. The caption is confused, authors have to delete the phrase “According to their knowledge of the product, the panelists had to decide whether it was a typical Gwell or not from an organoleptic point of view.” you have already explained in what consists step 1.
Figure 3: the caption is too dispersive; please try to be more concise explaining the data in figure and not the methodology or/and results
Line 190: So once again, an unexplained acronym. What’s mean OTU?
Author Response
Response to Reviewer 2 Comments
First of all, the authors would like to thank Reviewer 2 for the time dedicated to our manuscript and his constructive comments.
The yellow highlighted line numbers refer to the line number in the revised version of the manuscript.
General comments:
The manuscript titled “Microbial diversity associated with Gwell, a traditional French mesophilic fermented milk inoculated with a natural starter” has been focused on the relation between the Gwell organoleptic characteristics and the microbial community. The manuscript is interesting, it opened up a niche product (Gwell fermented milk), any way it results very useful in the experimental design structure that can be applied to other types of fermented foods. The introduction is well written, the results and discussion is too confused, it is probably due to the English language.
Concluding, I suggest the manuscript publication after minor revision and an in-depth English mother tongue revision.
Suggested corrections
My first doubt concerns the “Materials and Methods” section that it’s arranged at the end of the manuscript, why? Is there a reason, or is it a mistake?
> You are right, this is a mistake. This was amended, the ‘’Material and methods” section was put back just after the introduction section.
Line 377: please insert the bibliographic reference (i.e. R Core Team. R: A Language and Environment for Statistical Computing. Available online: https: //www.r-project.org (accessed on…dd mm year).
> Done, a reference was added using the citation() function from base R. : R Core Team. R: A Language and Environment for Statistical Computing. R Foundation for Statistical Computing, Vienna, Austria, 2020., Line 133.
Line 385: again, you have to insert the software references.
> Done, references were added using the citation() function from base R :
FactoMineR : Lê, S.; Josse, J.; Husson, F. FactoMineR: A Package for Multivariate Analysis. Journal of Statistical Software 2008, 25, 1–18. doi:10.18637/jss.v025.i01. Line 133.
and factoextra : Kassambara, A.; Mundt, F. factoextra: Extract and Visualize the Results of Multivariate Data Analyses, 2019. R package version 1.0.6. Line 138.
Line 393: authors have to specify the meaning of acronym ETS.
> Done, ETS is the french acronym for Salt Tryptone Water, the acronym was eliminated from the text and the exact composition of the Salt Tryptone Water specified under brackets. Line 144.
Line 397: please wright well the reagent name i.e. Anaerocult® A. Anyone could be able to reproduce your methods in the lab.
> Done, this was amended. We assumed the problem was the absence of the spacing between ® A and corrected this typo. Line 148
Line 419: is the “APILAB Plus computer software (API-bioMerieux, Basingstoke, UK)”, please reports all the references.
> Done, this was amended Line 171
Line 502: please insert also the reference for Illumina MiSeq
> Done, a reference was added (Ravi, R.K.; Walton, K.; Khosroheidari, M. MiSeq: A Next Generation Sequencing Platform for Genomic Analysis. In Disease Gene Identification: Methods and Protocols; DiStefano, J.K., Ed.; Methods in Molecular Biology, Springer: New York, NY, 2018; pp. 223–232. doi:10.1007/978-1-4939-7471-9_12.), Line 255
Paragraph 4.7: Authors must delete this paragraph. It is away off subject; you have inserted these considerations in the acknowledgement, it’s enough
> We recognize that the paragraph 4.7 can be perceived as out of the scope, but we would like to maintain it. Its objective is not to thank the Gwell producers but to explain the participatory research approach and the interactions between scientists and producers for the elaboration of the experimentation. To our knowledge, the participatory research approach in the field of microbiology is not yet really common.
Lines 89-94: the sentence is not clear, and the authors must rewrite it.
> This was amended. The corresponding section was rewritten in order to clarify it. We dissociated the categorization of the panelists from the categorization of the Gwell samples. The description of the results was standardized in three sentences based on the same pattern. Lines 285-289
Figure 1: The data have not scientific value due to the absence of error bars. The caption is confused, authors have to delete the phrase “According to their knowledge of the product, the panelists had to decide whether it was a typical Gwell or not from an organoleptic point of view.” you have already explained in what consists step 1.
> We reformulated the caption to only give information about the data in the figure. We can't provide error bars as it would have required multiple sensory evaluation sessions. However, the experimental setup allows us to get information on the robustness of the panelists judgments, as each sample was tasted in duplicate. This information wasn’t shown in figure 1, and it could indeed question the scientific value of the data. We amended figure 1, by distinguishing in each bar plot the judgments that were equal or different for each duplicated sample.
Figure 3: the caption is too dispersive; please try to be more concise explaining the data in figure and not the methodology or/and results
> Done, the caption was amended by removing the part explaining the methodology. Fig. 3
Line 190: So once again, an unexplained acronym. What’s mean OTU?
> Done, the acronym OTU for Operational Taxonomic Unit was detailed in the text, at its first appearance in the material and methods and in the results parts. Lines 262 & 385

Reviewer 3 Report
The paper can be considered for publication after following revisions.
- Authors need to point out the differences in process(s), if any, between Gwell, Villi, Dahi and Dadih in different countries, how Gwell practiced in France is different from those in other countries.
- Is the Gwell quality/problems mostly dependent on slowing down of acidification and texture or taste defects or it comes more from Milk, is there any possibility the culture evolves from batch to batch that impacts on the quality.
- The differences in bacterial ecosystem of the culture is it more regional if so, where there any cross studies done in the past in detail to analyze differences in the eco system.
- Even though the authors explain what this study/paper is about, the objective is not clear and why the study was conducted, is the study significant? Authors need to clearly define objectives
- A separate short section of Materials and Methods need to be outlined.
- Fig 5. The acidification activity of these strains seems to be poor. Do authors have any comparison data from any other reported studies and what is the acidification rate reported in Gwell, it is good to have the acidification rate in Gwell as positive control.
Author Response
Response to Reviewer 3 Comments
First of all, the authors would like to thank Reviewer 2 for the time dedicated to our manuscript and his constructive comments.
The yellow highlighted line numbers refer to the line number in the revised version of the manuscript.
The paper can be considered for publication after following revisions.
- Authors need to point out the differences in process(s), if any, between Gwell, Villi, Dahi and Dadih in different countries, how Gwell practiced in France is different from those in other countries.
> Viili, Dahi, Dadhi have very similar process compared to Gwell. As Gwell, they are obtained after the thermisation of the milk, by inoculation of a previous production, and incubation for a determined period at a determined temperature. The most important variations between these different products concern the temperature of the thermization step, the origin of the milk according to the considered country (cow, buffalo or even yak) and the duration and temperature of incubation during the fermentation step. As Dahi and Dadhi are mainly produced at domestic scale, the detailed process is very hard to determine. In general, the incubation is performed at room temperature, the milk is boiled from several minutes to one hour, then cooled before the inoculation step. For Viili, semi industrial scale production mentions a pasteurization step and an incubation step of 20h at 20°C. These informations were added in the introduction section with the corresponding references (Bakry AM and Campelo PH (2018). Mini-Review on Functional Characteristics of Viili and Manufacturing Process. J Food Biotechnol Res. Vol.2 No.1:7; Akuzawa R., Miura T., Surono I (2011). Asian fermented milks Encycl. Dairy Sci., pp. 507-511) Lines 61-69
- Is the Gwell quality/problems mostly dependent on slowing down of acidification and texture or taste defects or it comes more from Milk, is there any possibility the culture evolves from batch to batch that impacts on the quality.
> As the milk is thermized in Gwell production process, its impact on the quality of the final product would principally come from the physicochemical variations of its composition. Milk composition is globally stable with minor variations according to the season. These seasonal variations alone are too weak to explain Gwell defects or the loss of the starter. That is why our starting point is that the microbial composition of the starter can affect the acidification rate during the process and/or texture and taste of the final product. As you underlined it, the evolution of the starter microbial composition from batch to batch plays a major role on the quality of the Gwell, the starter evolution being the result of contaminations and/or of its intrinsic evolution.
- The differences in bacterial ecosystem of the culture is it more regional if so, where there any cross studies done in the past in detail to analyze differences in the eco system.
> You are right, the bacterial ecosystem composition could be influenced by geographical location even if the Gwell production area is quite small. It is one of the questions that we will like to explore. Unfortunately considering the weak number of Gwell producers, the study of the spatial structuration on the gwell will not be statistically representative.
- Even though the authors explain what this study/paper is about, the objective is not clear and why the study was conducted, is the study significant? Authors need to clearly define objectives
> Our work has two main objectives. The first one, the applied objective, is to help gwell producers to better understand their ecosystem in order to better control it and to avoid gwell loss accident (economic prejudice). The second one, more academic, is to characterize in detail the microbial community and how it is structured. This descriptive study sets the base to analyze the factors influencing Gwell microbial composition and to understand the interactions between the different strains in the microbial community. The introduction section was amended in order to clarify this point. Line 83-85
- A separate short section of Materials and Methods need to be outlined.
> In the first version, the material and method section was put after the results section by mistake. This was amended in the revised version.
- Fig 5. The acidification activity of these strains seems to be poor. Do authors have any comparison data from any other reported studies and what is the acidification rate reported in Gwell, it is good to have the acidification rate in Gwell as positive control.
> In this experiment only singles strains were considered and you are right, in these conditions the acidification rate is relatively weak compared to the acidification rate of the gwell where several of these strains are mixed. On average, the acidification rate of Gwell samples reaches pH 4.7 in 4 hours. In order to illustrate it, and as you proposed, we added in the figure 5 the acidification curves of three gwell samples as positive controls and the material and methods, paragraph 2.5.7. (Acidifying capacity of strains) amended to introduce these controls. Lines 221-222.